# A Predictive Model for the Development of Long COVID in Children

**DOI:** 10.3390/ijerph22111693

**Published:** 2025-11-09

**Authors:** Vita Perestiuk, Andriy Sverstyuk, Tetyana Kosovska, Liubov Volianska, Oksana Boyarchuk

**Affiliations:** 1Department of Children’s Diseases and Pediatric Surgery, I. Horbachevsky Ternopil National Medical University, 46001 Ternopil, Ukraine; kosovska@tdmu.edu.ua (T.K.); volyanska@tdmu.edu.ua (L.V.); boyarchuk@tdmu.edu.ua (O.B.); 2Department of Medical Informatics, I. Horbachevsky Ternopil National Medical University, 46001 Ternopil, Ukraine; sverstyuk@tdmu.edu.ua

**Keywords:** COVID-19, long COVID, post-COVID syndrome, children, adolescents, mathematical model, forecasting

## Abstract

Background/Objectives: Machine learning is an extremely important issue, considering the potential to prevent the onset of long-term complications from coronavirus disease or to ensure timely detection and effective treatment. The aim of our study was to develop an algorithm and mathematical model to predict the risk of developing long COVID in children who have had acute SARS-CoV-2 viral infection, taking into account a wide range of demographic, clinical, and laboratory parameters. Methods: We conducted a cross-sectional study involving 305 pediatric patients aged from 1 month to 18 years who had recovered from acute SARS-CoV-2 infection. To perform a detailed analysis of the factors influencing the development of long-term consequences of coronavirus disease in children, two models were created. The first model included basic demographic and clinical characteristics of the acute SARS-CoV-2 infection, as well as serum levels of vitamin D and zinc for all patients from both groups. The second model, in addition to the aforementioned parameters, also incorporated laboratory test results and included only hospitalized patients. Results: Among 265 children, 138 patients (52.0%) developed long COVID, and the remaining 127 (48.0%) fully recovered. We included 36 risk factors of developing long COVID in children (DLCC) in model 1, including non-hospitalized patients, and 58 predictors in model 2, excluding them. These included demographic characteristics of the children, major comorbid conditions, main symptoms and course of acute SARS-CoV-2 infection, and main parameters of complete blood count and coagulation profile. In the first model, which accounted for non-hospitalized patients, multivariate regression analysis identified obesity, a history of allergic disorders, and serum vitamin D deficiency as significant predictors of long COVID development. In the second model, limited to hospitalized patients, significant risk factors for long-term sequelae of acute SARS-CoV-2 infection included fever and the presence of ≥3 symptoms during the acute phase, a history of allergic conditions, thrombocytosis, neutrophilia, and altered prothrombin time, as determined by multivariate regression analysis. To assess the acceptability of the model as a whole, an ANOVA analysis was performed. Based on this method, it can be concluded that the model for predicting the risk of developing long COVID in children is highly acceptable, since the significance level is *p* < 0.001, and the model itself will perform better than a simple prediction using average values. Conclusions: The results of multivariate regression analysis demonstrated that the presence of a burdened comorbid background—specifically obesity and allergic pathology—fever during the acute phase of the disease or the presence of three or more symptoms, as well as laboratory abnormalities including thrombocytosis, neutrophilia, alterations in prothrombin time (either shortened or prolonged), and reduced serum vitamin D levels, are predictors of long COVID development among pediatric patients.

## 1. Introduction

Long-term follow-up studies of patients who had experienced severe acute respiratory syndrome caused by SARS-CoV-2 revealed a broad spectrum of persistent health abnormalities in approximately one in ten individuals. These manifestations have been collectively defined as the clinical syndrome of long COVID. To date, the growing body of scientific literature reports at least 65 million cases of the disease worldwide, affecting both adult and pediatric populations [1].

Long COVID represents a substantial burden on public health due to the ongoing circulation of the virus within the human population and its rapid genetic variability [2]. The disease may occur even after mild or asymptomatic initial illness and is characterized by a highly heterogeneous and diverse clinical presentation, encompassing over 200 reported symptoms [3,4]. Importantly, post-COVID-19 syndrome has a substantial impact on individuals’ physical and mental health, work capacity, professional functioning, and overall quality of life, regardless of age, constituting a serious global public health problem [5,6,7].

Numerous studies from various regions have documented the burden of long COVID in adults over a one-year follow-up period, identifying key symptoms, associated outcomes, risk factors, and potential pathophysiological mechanisms [8,9]. However, there remains a lack of consensus regarding the definition of post-COVID syndrome in children, particularly in terms of the type, number, and duration of clinical manifestations, as well as the mechanisms underlying its development. The absence of pediatric-specific studies, standardized case definitions, and sufficient research investment further complicates the understanding and management of this condition in the pediatric population [10].

Current data on the prevalence of long COVID in children and adolescents are highly variable, ranging from 0% to 70% across different studies [11,12]. However, it is important to note that significant heterogeneity exists among study cohorts with respect to the SARS-CoV-2 variants, clinical manifestations of COVID-19, age distribution, the presence or absence of comorbidities, COVID-19 vaccination status, country-specific factors, and follow-up durations, all of which may influence the presentation and outcomes of long COVID [13]. According to the World Health Organization (WHO), approximately 10–20% of individuals who experience an acute SARS-CoV-2 infection develop long COVID [14]. It is evident that the study of long COVID clinical features and their prevalence in children requires further research, as the high rates of acute COVID-19 infections remain substantial and continue to present a significant challenge to public health systems [15,16].

Understanding the burden of disease and identifying populations at increased risk of developing long-term complications can inform targeted preventive strategies and ensure effective support for affected individuals, thereby improving long COVID outcomes and reducing health disparities resulting from inadequate rehabilitation and recovery services [17].

In the era of computerization and the widespread implementation of artificial intelligence, it is reasonable to leverage their capabilities and the potential of multifactorial analysis in studying complex pathological conditions such as long COVID [18]. Several models have been published for stratifying the risk of long COVID among adult patients who have experienced acute infection. For instance, Sudre et al. [19] demonstrated that individuals—independent of age or sex—who reported more than five symptoms during the first week of illness (with fatigue, headache, dyspnea, hoarseness, and myalgia being the most prominent) exhibited a substantially increased risk of developing long COVID. Nonetheless, their study was based on a non-representative cohort composed exclusively of mobile application users, thereby limiting the external validity of the findings. Similarly, Antony et al. [20] identified age, sex, obesity, hypertension, diabetes mellitus, and chronic pulmonary disease as significant predictors of long COVID; however, their predictive model did not incorporate laboratory-based parameters.

Predictive models specific to pediatric populations remain limited. Several studies have demonstrated that age above 10–12 years—where each additional year is associated with an 8% increase in risk [21]—as well as muscle pain during hospitalization and admission to the intensive care unit (ICU), are significantly associated with the development of long COVID in children and adolescents [22,23]. Additional research indicates that comorbidities, particularly allergic diseases, the presence of acute-phase symptoms such as chest pain, dyspnea, and anosmia or ageusia, along with their severity, hospitalization during the acute SARS-CoV-2 infection, and infection with the Alpha or Delta variants, are linked to an increased likelihood of post-COVID syndrome [24,25]. In a multivariate analysis conducted by Trapani et al. [26], advancing age was also identified as a significant risk factor for long COVID in pediatric patients, whereas sex and pre-existing conditions showed no association with symptom persistence.

The long-term consequences of SARS-CoV-2 infection in children have implications that extend beyond individual morbidity, posing significant challenges for healthcare systems and public health planning. Early recognition of individuals at risk for long COVID could facilitate timely intervention, optimize rehabilitation resources, and reduce the social and economic burden associated with prolonged symptoms.

However, despite the growing recognition of long COVID as a major public health concern, practical tools for early risk stratification in pediatric populations are lacking. Advances in artificial intelligence and data-driven modeling offer a promising opportunity to integrate multiple clinical, demographic, and laboratory variables to predict long COVID risk with greater precision. Such approaches could support clinicians and policymakers in developing personalized monitoring and preventive strategies for affected children.

Given the predominance of predictive models developed for adult cohorts, the aim of the present study was to develop an algorithm and mathematical model capable of estimating the risk of long COVID in pediatric patients following acute SARS-CoV-2 infection, incorporating a broad spectrum of demographic, clinical, and laboratory parameters.

## 2. Materials and Methods

A cross-sectional study was carried out involving 305 children aged between 1 month and 18 years who had recovered from acute SARS-CoV-2 infection. Hospitalized participants were treated in the pediatric infectious diseases department at “Ternopil Municipal City Hospital No. 2” between September 2022 and May 2024. Other participants sought pediatric consultation in Ternopil due to symptoms of acute infection or long-term COVID-19 sequelae.

The inclusion criteria comprised: (1) children aged between 1 month and 18 years; (2) laboratory-confirmed SARS-CoV-2 infection, verified by polymerase chain reaction (PCR), rapid antigen testing, or serological detection of IgM antibodies, applied interchangeably; and (3) written consent for participation provided by a parent or legal guardian.

The exclusion criteria included: (1) absence of laboratory confirmation of COVID-19 by any diagnostic method; (2) refusal of the family to participate in subsequent follow-up assessments; and (3) lack of valid contact information or inability to establish communication with the family.

All procedures adhered to the ethical principles outlined in the Helsinki Declaration of 1975 (revised in 2000). The study protocol received approval from the Bioethics Committee of the I. Ya. Horbachevsky Ternopil National Medical University, Ministry of Health of Ukraine (protocol No. 70, dated 1 August 2022). Informed voluntary consent was obtained from all parents or from participants aged 16 and above prior to inclusion in the study, and for use of diagnostic and treatment data for research purposes.

Comprehensive data collection included demographic characteristics (age at diagnosis, sex), comorbidities, and key clinical features of COVID-19, such as symptoms, disease course, severity, and length of hospitalization. Allergic pathology included food allergy, allergic rhinitis, and atopic dermatitis. Nutritional disorders included overweight, obesity, and malnutrition. Laboratory test results were analyzed for hospitalized patients.

COVID-19 severity was classified according to WHO guidelines, distinguishing asymptomatic, mild, moderate, severe, and critical cases [27].

Serum 25-hydroxyvitamin D (25(OH)D) levels were measured by ELISA using FCCu Bind ELISA Microwells (ARP American Research Products, Inc., Waltham, MA, USA). Based on recommendations from the European Vitamin D Association (EVIDAS) (Warsaw, Poland), optimal vitamin D status was defined as 30–100 ng/mL (75–250 nmol/L), insufficiency as 20–30 ng/mL (50–75 nmol/L), and deficiency as <20 ng/mL (<50 nmol/L) [28].

Serum zinc concentration was determined using a Multiskan FC-357 microplate photometer (Thermo Fisher Scientific, Waltham, MA, USA) with a colorimetric zinc assay kit (Elabscience E-BC-K137-M, Houston, TX, USA). Children were categorized into groups with low or adequate serum zinc levels. Since no universally accepted threshold exists, a cut-off of 10.7 μmol/L (<70 µg/dL) was applied to define clinical zinc deficiency [29].

All participants underwent follow-up monitoring during the post-COVID period to identify long COVID symptoms. Assessments were conducted in person or by phone at intervals of 1–3, 3–6, 6–9, and 9–12 months after acute infection, utilizing the ISARIC/IP4C Global Pediatric COVID-19 Follow-up Form developed by the International Severe Acute Respiratory and Emerging Infection Consortium (ISARIC).

Participation and follow-up were voluntary. Questionnaires were completed by children aged 8 and older or by their parents if younger. Long COVID was defined according to WHO criteria as symptoms persisting or emerging at least three months post-infection and lasting for a minimum of two months without alternative explanations [30]. Children without ongoing symptoms by eight weeks post-acute infection were considered fully recovered.

Out of 305 enrolled children, 40 declined further follow-up. Among those who continued, 187 were hospitalized patients, while 78 formed the outpatient group consulting pediatricians for acute or post-COVID symptoms.

To analyze factors influencing long COVID development, two predictive models were developed. The first included basic demographic and clinical data, plus vitamin D and zinc serum levels for all patients. The second model incorporated these variables alongside laboratory test results and included only hospitalized patients.

Stepwise multivariate regression analysis assessed the significance of potential predictors. Initial correlation matrices ensured absence of multicollinearity (r > 0.7) among risk factors. Regression coefficients (“B”) quantified the impact of each predictor on the probability of long COVID. Variables with *p* > 0.05 were excluded iteratively.

The first model identified 4 significant predictors out of 36 variables, while the second revealed 7 significant predictors from 58 variables. Subsequent correlation analyses confirmed the absence of multicollinearity among these predictors. Two final multivariate regression models were built to estimate long COVID risk.

The model’s performance was evaluated using Nagelkerke’s R^2^ coefficient, and its adequacy was confirmed through analysis of variance (ANOVA) performed with the StatSoft STATISTICA version 10.0 (Informer Technologies, Inc., Los Angeles, CA, USA).

## 3. Results

Among 265 children, 138 patients (52.0%) developed long COVID, and the remaining 127 (48.0%) fully recovered.

The mean duration of follow-up from the time of hospitalization was 10.9 ± 2.0 months. Follow-up assessments were completed by 127 participants at 1–3 months, 125 at 3–6 months, 122 at 6–9 months, and 101 at 9–12 months post-infection.

The most frequently reported symptoms included fatigue (52.0%), reduced physical activity (44.1%), and headache (35.3%). By 12 months, the prevalence of these symptoms had decreased to 19.8%, 13.9%, and 12.3%, respectively (*p* < 0.00001 for all trends).

When symptoms were grouped by physiological system, neurological manifestations were reported by 64.6% of participants, followed by general symptoms (59.8%), musculoskeletal (15.0%), gastrointestinal (11.8%), cardiac (7.9%), respiratory (7.1%), sensory (8.7%), and balance-related (6.6%) symptoms. Notably, 62.2% of children experienced manifestations affecting more than one organ system.

Among neurological symptoms, the prevalence of insomnia (21.3% → 1.0%), memory impairment (23.5% → 8.8%), emotional excitability (25.2% → 1.0%), and concentration difficulties (22.7% → 0%) declined significantly throughout the observation period (*p* < 0.0001). Musculoskeletal complaints, primarily muscle pain, also decreased markedly (27.9% → 10.5%; *p* < 0.00001).

At 6 months post-COVID-19, 14.0% of children had fully recovered; this proportion increased to 43.9% at 9 months and 67.5% at 12 months. Nevertheless, by the end of the 12-month follow-up, 32.5% of patients continued to experience at least one long COVID symptom. The most persistent manifestations included fatigue (19.8%), reduced physical activity (13.9%), and headache (12.3%), followed by muscle pain (10.5%), memory impairment (8.8%), and lack of energy (8.0%).

We included 36 risk factors of developing long COVID in children (DLCC) in model 1 (Table 1), including non-hospitalized patients, and 58 predictors in model 2, excluding them (Table 2). These included demographic characteristics of the children, major comorbid conditions, main symptoms and course of acute SARS-CoV-2 infection, and main parameters of complete blood count and coagulation profile.

The results of multivariate regression analysis were taken as mathematical models, which allows, based on the data of regression coefficients “B” and the values of predictors that have a significant impact on the development of the risk of post-COVID syndrome, to predict the probability of its occurrence in children. Two multivariate regression models were constructed, taking into account only 4 and 7 reliable factors (Table 3 and Table 4). Based on the results obtained, which are presented in Table 3 and Table 4, mathematical models were created to determine the risk factors for the DLCC.

Based on the results obtained, which are presented in Table 3 and Table 4, we built mathematical models to determine a risk coefficient for developing long COVID in children (RCDLCC):RCDLCC 1 = 0.4266 × X26 + 0.2855 × X28 + 0.0041 × X33 + 0.3972 × X34 + 0.0066RCDLCC 2 = 0.3018 × X5 + 0.1579 × X15 + 0.0600 × X24 + 0.3383 × X30 − 0.0307 × X32 + 0.4646 × X37 + 0.1791 × X47 − 0.0644
where X5, X15, X24, X26, X28, X30, X32, X33, X34, X37, X47—significant predictors of RCDLCC with regression coefficients “B”.

We also evaluated the residual deviations to analyze the quality of multivariate regression models 1 and 2 (Figure 1a,b). As shown in the histograms, the residual deviations are distributed symmetrically in both cases, approaching the normal distribution curve of the residuals, so the statistical hypothesis that their distribution corresponds to the normal distribution law is not rejected.

A normal probability plot was constructed to confirm the normal distribution of the residual deviations (Figure 2). The plot data indicate the absence of systematic deviations from the normal probability line. This indicates that the residual deviations are distributed according to the normal distribution law.

To assess the acceptability of the model as a whole, an ANOVA analysis was performed (Table 5 and Table 6). Based on this method, it can be concluded that the model for predicting the risk of developing prolonged COVID in children is highly acceptable, since the significance level is *p* < 0.001, and the model itself will perform better than a simple prediction using average values.

For additional assessment of the quality of the mathematical model of the risk of DLCC, the Najdelkirk determination coefficient (R^2^) was analyzed, which shows what part of the factors is taken into account in the prediction. In the proposed mathematical model of the risk of DLCC taking into account outpatients, the determination coefficient is R^2^ = 0.971 (in the Statistica 10.0 program, R^2^ = 0.9714339). Thus, 97.1% of the factors were taken into account. In model 2, without taking into account outpatients, the determination coefficient is R^2^ = 0.964 (in the Statistica 10.0 program, R^2^ = 0.9641852). That is, 96.4% of the factors were taken into account in the risk of DLCC prediction model. The determination coefficient indicates how much the obtained observations confirm the mathematical model.

## 4. Discussion

In the present study, we investigated demographic characteristics, comorbidities, acute-phase clinical manifestations of SARS-CoV-2 infection, and laboratory parameters as potential risk factors associated with the development of long COVID in the pediatric population.

The results of multivariate regression analysis showed that the presence of a burdened comorbid background, as well as obesity and allergic pathology, are predictors of the development of long COVID among pediatric patients. Comparable results were observed in our previous investigation of the coagulation profile in pediatric patients, further supporting the validity and robustness of the current findings [31].

Similar results have been obtained by other researchers. In particular, Wongwathanavikrom et al. [32] described that allergic diseases are important in the development of post-COVID syndrome. This observation is in agreement with our results. Tsilingiris et al. [33] are among the first to study in detail a wide range of clinical prognostic factors of long COVID. They indicate that the main risk factors that increase the likelihood of developing the disease are certain comorbidities, including obesity, type 2 diabetes, connective tissue diseases, allergic rhinitis. In a large study by Antony et al. [20] involving more than 2 million patients, it was indicated that comorbid conditions such as obesity, hypertension, diabetes mellitus and chronic lung disease were predictors of the development of long-term consequences of COVID-19. These findings are consistent with our results, given that no cases of diabetes mellitus or chronic lung disease were observed among the participants. Researchers from the United States used a superlearning algorithm to develop and validate models for predicting the risk of long COVID. In models that used phenotype data from the period before acute SARS-CoV-2 infection, overweight, obesity, and circulatory diseases, including hypertension, were identified as diagnoses predictive of post-COVID syndrome [34]. Maddux et al. [35] showed that patients with respiratory diseases, namely bronchial asthma, or obesity have a higher risk of prolonged recovery. The reported data are consistent with the findings presented in our study.

Prognostic models of other researchers differ. A large cohort study in China was conducted to assess the risk of developing long-term outcomes of COVID-19 after the Omicron wave. The researchers performed univariate regression analysis and showed that history of diabetes, chronic kidney and liver disease were predictors of long COVID [36]. Researchers from the United States found that severe primary disease caused by the SARS-CoV-2 virus, underweight and the presence of comorbidities at the beginning of the study (e.g., cancer and cirrhosis) are likely associated with an increased risk of developing long COVID [37]. According to Subramanian et al. [8], comorbidities were significantly associated with the development of post-COVID syndrome, with the strongest associations observed for chronic obstructive pulmonary disease (COPD), depression, and celiac disease.

Our analysis demonstrated that the presence of fever during the acute phase of infection and the occurrence of three or more symptoms in total were associated with an increased likelihood of developing long COVID. The findings of other researchers are also inconsistent. Several scientists noted that the presence of symptomatic acute SARS-CoV-2 infection or the number of symptoms of 4 or more increases the likelihood of developing long COVID in children [21,23,38,39]. In contrast, Jin et al. [34] reported that during the initial illness, the prognostic signs and symptoms were predominantly respiratory in nature, including shortness of breath and respiratory distress.

We found that an increased likelihood of developing long COVID in pediatric patients was also associated with specific alterations in laboratory parameters, most notably thrombocytosis, neutrophilia, and deviations in prothrombin time, including both shortened and prolonged values. Fang et al. [36] also determined that elevated interleukin-6 (IL-6) and procalcitonin, leukopenia, lymphopenia, eosinophilia, neutrophilia, thrombocytosis, increased D-dimer, activated partial thromboplastin time (aPTT), thrombin time, and prothrombin time were predictors of long COVID. These results are consistent with our data.

Other researchers have reported opposite findings. For instance, Tsilingiris et al. [33] conducted a large-scale investigation aimed at identifying a range of laboratory parameters, the alterations of which were associated with an increased risk of developing long-term sequelae of COVID-19. In particular, the cardinal laboratory parameters with prognostic potential are increased IL-6, C-reactive protein (CRP), D-dimer, endothelin-1 (ET-1), decreased angiopoietin-2 (Ang-2) and cortisol, detection of SARS-CoV-2 RNA in feces or intestinal mucosa, biomarkers of Epstein-Barr virus reactivation, anti-IFN-α2 or anti-IFN-λAbs, metabolites of mitochondrial dysfunction, increased percentage of cerebrospinal fluid abnormalities, and increased biomarkers of neuronal damage. Their results markedly differ from ours, as most of the parameters they analyzed were not included in our investigation. Wang et al. [40] developed machine learning models using multiplex cytokines, proteins, and metabolites. Network analysis showed that within the minimum prediction panel, there was a decrease in the metabolites spermidine and taurine, accompanied by a decrease in the concentrations of protective cytokines (IL-22 and colony-stimulating factor 3 (CSF 3) and an increase in proinflammatory cytokines (IL-15 with a simultaneous increase in IL-10. Jin et al. [34] further indicated that the three most frequently observed laboratory parameters associated with long COVID were hematocrit, hemoglobin, and red blood cell concentration; however, these associations were not statistically significant. Other researchers have demonstrated that elevated levels of specific inflammatory markers present during the acute phase of infection—namely, CRP, IL-6, and tumor necrosis factor (TNF)—are correlated with an increased risk of developing long COVID [41].

We found that low 25-hydroxyvitamin D (25(OH)D) levels were a risk factor for post-COVID syndrome. This finding has been confirmed by many studies. For example, lower 25(OH)D concentrations were reported six months after the acute phase of illness in adults with long-term COVID compared with those who fully recovered from COVID-19 infection (20.1 vs. 23.2 ng/mL, *p* = 0.0300) [42]. Similarly, Chen et al. [43] demonstrated that vitamin D deficiency was associated with delayed recovery in adults with long COVID. A study by Guerrero-Romero et al. [44] found a threefold increased risk of developing long COVID in adults with insufficient vitamin D and magnesium levels. Our other work also found that children with 25(OH)D deficiency or insufficiency were 2.2 times more likely to develop long-term sequelae of COVID-19 compared with those with optimal vitamin D levels [45]. However, several studies in adults have not found a significant association between low serum vitamin D levels and post-COVID syndrome [46,47,48].

Among the key predictors identified in pediatric populations that influence the development of long COVID are older age and female sex. However, in our study, these factors were not established as statistically significant risk factors. Adler et al. [21] reported that each additional year of age increased the risk by approximately 8%. Similarly, Atchison et al. [23] found that older children were three times more likely to report persistent symptoms compared to younger ones. The logistic regression model developed by Seery et al. [38] also demonstrated a significant association between older age and the persistence of symptoms lasting at least three months after acute COVID-19. Consistent with these findings, Camporesi et al. [49] identified age above 12 years and female sex as statistically significant predictors of post-COVID syndrome. Moreover, a recent meta-analysis further confirmed the association between older age (above 10 years) and an increased risk of developing long COVID [50].

The present study did not assess the influence of specific pharmacological agents—such as antiviral or antibacterial medications, anticoagulants, or corticosteroids—on the development of post-COVID-19 syndrome in the pediatric population. This decision was based on the relatively small sample size and the greater clinical relevance of these treatments in adult cohorts. Findings from studies involving adult populations are generally consistent with our results. Researchers from Italy conducted a retrospective study involving outpatients and inpatients. According to their multivariate analysis, treatment with glucocorticoids during acute infection (in the presence of pneumonia complicated by respiratory failure) was identified as a prognostic factor for the development of long COVID [51]. A multivariable analysis conducted by researchers in the Netherlands demonstrated that corticosteroid treatment in patients hospitalized with COVID-19 was associated with a significantly lower likelihood of developing post-COVID syndrome (OR 0.32, 95% CI 0.11–0.90, *p* = 0.03). In contrast, treatment with antibacterial agents was associated with a higher likelihood (OR 1.26, 95% CI 0.47–3.39, *p* = 0.65), although this difference did not reach statistical significance [52].

Long COVID constitutes a substantial burden for both pediatric patients and public health systems. Accordingly, it is imperative to provide education to caregivers and children regarding this condition beyond the acute phase, even in cases where the initial SARS-CoV-2 infection was asymptomatic [53]. Standardized instruments for the diagnosis and monitoring of long COVID in the pediatric population have been developed. It is recommended that all children with a history of SARS-CoV-2 infection undergo assessment at 1 and/or 3 months following the acute phase to identify any persistent symptoms that may interfere with daily functioning [54]. Nevertheless, a comprehensive medical history and meticulous physical examination following SARS-CoV-2 infection remain fundamental components of the diagnostic process. In the majority of cases, a symptom-oriented assessment is sufficient, whereas only a limited subset of patients requires further diagnostic evaluation [55].

For children presenting with persistent symptoms, an initial assessment that includes routine blood tests (complete blood count, liver and renal function tests, glucose levels, and coagulation profile), electrocardiography, lung ultrasound, and the exclusion of other identifiable and treatable conditions with similar clinical manifestations (such as hypothyroidism, autoimmune disorders, or celiac disease) represents a rational and evidence-based approach [54]. In brief, children exhibiting persistent symptoms three months after infection should first undergo basic routine testing to exclude other treatable conditions. If initial results are normal, a tailored second- and third-level multidisciplinary evaluation should be conducted, guided by the child’s predominant symptoms [56].

The management of pediatric long COVID necessitates an interdisciplinary approach encompassing thorough clinical assessment, targeted symptom management, and provision of psychosocial support [55].

An essential initial step involves providing families with transparent information regarding both the established and uncertain aspects of long COVID. This includes emphasizing the importance of mutual trust, continuous engagement with emerging research, and collaboration with local, national, or international networks dedicated to pediatric post-COVID conditions for the exchange of experiences and preliminary data [54].

Ideally, all affected children should be enrolled in clinical studies aimed at elucidating the burden, duration, social impact, phenotypic characteristics, and diagnostic approaches of pediatric long COVID [54].

In children, routine activities are often disrupted in the context of long COVID. Providing formal documentation (or establishing direct communication) for school personnel, sports instructors, and close relatives or peers can facilitate individualized scheduling and enhance awareness among those in close contact with the child. Such measures may significantly support the child’s gradual reintegration into social environments and promote understanding and assistance from their immediate community [54].

Regardless of whether post-COVID-19 syndrome represents an organic disorder or has a psychological basis, mental health support should always be provided when the well-being of the child or family appears compromised. Given that children with long COVID often experience a sudden decline in their quality of life, psychological consequences are common and warrant careful attention and intervention, following the same principles applied to the management of any pediatric mental health concern.

To date, no therapeutic interventions have been identified that effectively prevent the development of PCC following SARS-CoV-2 infection. Evidence from adult studies indicates that COVID-19 vaccination can substantially reduce—but not completely eliminate—the risk of PCC after breakthrough infections [57]. Several studies involving pediatric populations have demonstrated that vaccination was associated with a reduced risk of developing long COVID at 3, 6, and 12 months among older children, as well as with a lower risk of reinfection [49,58,59]. Therefore, the most effective preventive strategy remains avoidance of SARS-CoV-2 infection. However, it is important to recognize that nonpharmacological interventions aimed at reducing viral transmission have had profound negative impacts on children’s mental and physical health [60]. Therefore, families and communities should strive to adopt balanced, evidence-based approaches that protect both infection control and the child’s overall well-being.

### Limitations of the Study

The present study has several limitations that should be acknowledged. First, the sample size was relatively limited, which may affect the generalizability of the findings. Second, a considerable proportion of participants were recruited from tertiary healthcare facilities, potentially introducing selection bias, as these patients may differ from those treated in primary or secondary care settings. Furthermore, as this study was conducted at a single center, its representativeness may be limited. The primary focus of our study was the development of the model; therefore, external validation was not conducted within the scope of this research. However, external validation is recommended prior to the clinical application of these models. In particular, it is essential to perform such validation across different settings and populations to ensure their generalizability and robustness. Another limitation is the absence of an assessment of potential discrepancies between child self-reported and parent-reported symptoms, as well as the lack of evaluation of inter-rater reliability between these reporting methods. This may have influenced the consistency of reports for subjective symptoms such as fatigue and emotional lability. Furthermore, both in-person and telephone follow-up interviews were employed, which could have introduced variability in responses. Nevertheless, to minimize methodological bias and ensure data consistency, the standardized ISARIC Paediatric COVID-19 questionnaire was used across both modes of data collection.

## 5. Conclusions

The algorithm and mathematical model proposed for the first time for predicting the risk of developing long COVID in children have high acceptability and quality. When conducting multivariate regression analysis, it was found that the predictors of the development of post-COVID syndrome are the presence of a history of obesity and allergic pathology, fever during the acute phase of the disease or a total number of symptoms of 3 or more, as well as laboratory parameters, in particular thrombocytosis, neutrophilia, changes in prothrombin time and a decrease in serum vitamin D levels.

The developed prognostic model will allow for timely identification and monitoring of pediatric patients with acute SARS-CoV-2 infection who have an increased likelihood of developing post-COVID syndrome. This will contribute to the creation of adapted prevention programs and improve the quality of life of children with long COVID.

## Figures and Tables

**Figure 1 ijerph-22-01693-f001:**
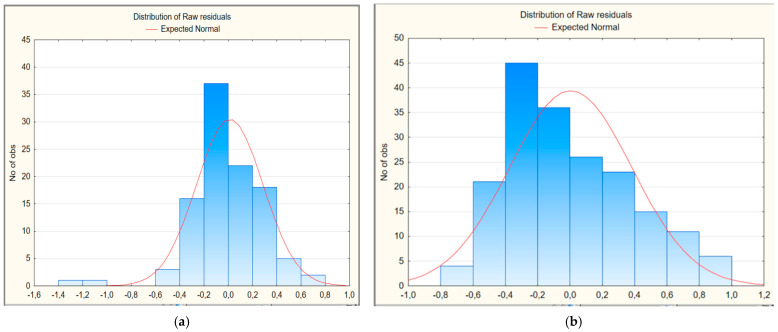
Histogram of residual deviations of multivariate regression models for predicting the risk of developing long COVID in children: (**a**) first model; (**b**) second model.

**Figure 2 ijerph-22-01693-f002:**
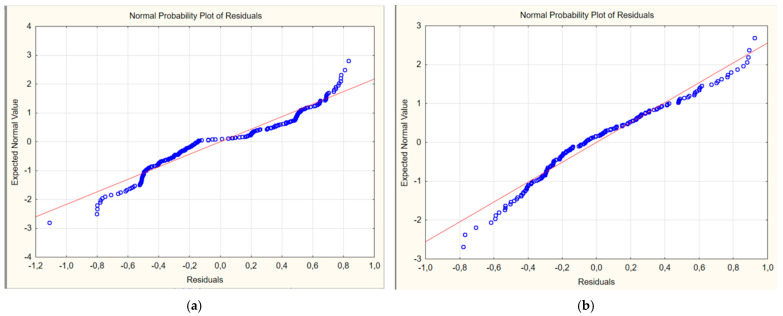
Normal probability plot of residual deviations of multivariate regression models for predicting the risk of developing long-term COVID in children: (**a**) first model; (**b**) second model.

**Table 1 ijerph-22-01693-t001:** Predictors that were analyzed to determine the risk of DLCC in model 1.

Conventional Designations of Factors in the Mathematical Forecasting Model	Predictor Name	Regression Coefficient (B)	Standard Error, SE (B)	Significance Level *p*
X1	Age	0.0147	0.0128	0.2491
X2	Under 6 years, over 6 years	0.1115	0.1307	0.3945
X3	Female	0.1428	0.0609	0.0199
X4	COVID-19 course	0.0754	0.0627	0.2300
X5	Fever	0.1917	0.0939	0.0422
X6	Runny nose	0.0143	0.0778	0.8542
X7	Sore throat	0.0756	0.1111	0.4970
X8	Hoarseness	−0.0624	0.1155	0.5894
X9	Vomiting	−0.1550	0.1015	0.1280
X10	Diarrhea	0.0447	0.1348	0.7405
X11	Cough	0.0736	0.0771	0.3410
X12	Shortness of breath	0.0541	0.1585	0.7331
X13	Weakness	−0.0930	0.0879	0.2910
X14	Loss of appetite	0.0190	0.0830	0.8190
X15	Number of COVID-19 symptoms	0.0130	0.0533	0.8074
X16	Number of symptoms ≥3	0.1903	0.1114	0.0887
X17	Number of symptoms ≥4	0.0384	0.1016	0.7061
X18	Neurological diseases	0.0625	0.1096	0.5691
X19	Gastrointestinal diseases	−0.0612	0.1433	0.6699
X20	Heart diseases	0.2045	0.1740	0.2410
X21	Allergic rhinitis	0.0692	0.1253	0.5816
X22	Food allergy	−0.0824	0.1372	0.5488
X23	Atopic dermatitis	−0.1502	0.1178	0.2033
X24	Kidney problems	−0.0258	0.1596	0.8715
X25	Overweight	0.4689	0.2162	0.0310
X26	Obesity	0.6111	0.1233	<0.0001
X27	Malnutrition	−0.1516	0.3061	0.6209
X28	Allergic pathology	0.2631	0.0923	0.0047
X29	Overweight/obesity	−0.5864	0.2648	0.0277
X30	Nutritional disorders	0.0444	0.0964	0.6454
X31	Comorbid diseases	0.1021	0.0827	0.2185
X32	Comorbid diseases ≥2	0.0382	0.0890	0.6685
X33	Vitamin D, ng/mL (N > 30)	0.0039	0.0018	0.0305
X34	Vitamin D deficiency	0.3515	0.0906	0.0001
X35	Zinc, μmol/L (Normal: 10.7–24.0)	0.0056	0.0050	0.2575
X36	Zinc deficiency	−0.1011	0.0728	0.1664

**Table 2 ijerph-22-01693-t002:** Predictors that were analyzed to determine the risk of DLCC in model 2.

Conventional Designations of Factors in the Mathematical Forecasting Model	Predictor Name	Regression Coefficient (B)	Standard Error, SE (B)	Significance Level *p*
X1	Age	0.0173	0.0182	0.3426
X2	Age under 6 years, over 6 years	−0.0154	0.1830	0.9330
X3	Female gender	0.1729	0.0718	0.0172
X4	COVID-19 course	0.2032	0.0838	0.0164
X5	Fever	0.4517	0.1484	0.0027
X6	Runny nose	0.0779	0.0819	0.3431
X7	Sore throat	0.1597	0.1447	0.2714
X8	Hoarseness	0.0712	0.1122	0.5263
X9	Vomiting	0.0678	0.1044	0.5170
X10	Diarrhea	0.1077	0.1423	0.4502
X11	Cough	−0.0486	0.0853	0.5695
X12	Shortness of breath	−0.0523	0.1997	0.7939
X13	Loss of appetite	0.0411	0.0940	0.6624
X14	Number of COVID-19 symptoms	−0.1115	0.0566	0.0505
X15	Number of symptoms ≥3	0.2800	0.1016	0.0065
X16	Number of symptoms ≥4	0.0769	0.1154	0.5065
X17	Neurological diseases	−0.0661	0.1190	0.5794
X18	Gastrointestinal diseases	−0.0013	0.1796	0.9941
X19	Heart diseases	0.0331	0.2498	0.8947
X20	Kidney problems	−0.0190	0.2071	0.9271
X21	Overweight	−0.1105	0.1251	0.3788
X22	Obesity	0.1464	0.1398	0.2965
X23	Malnutrition	0.1121	0.1097	0.3080
X24	Allergic pathology	0.4379	0.0714	<0.0001
X25	Overweight/obesity	−0.2896	0.1842	0.1177
X26	Nutritional disorders	0.1088	0.0989	0.2727
X27	Comorbid diseases	0.2336	0.0909	0.0110
X28	Comorbid diseases ≥2	0.0552	0.1045	0.5979
X29	Platelets	<−0.0001	0.0003	0.9076
X30	Thrombocytosis	0.4088	0.1070	0.0002
X31	Thrombocytopenia	0.0052	0.1448	0.9717
X32	Leukocytes	−0.0515	0.0190	0.0073
X33	Leukocytosis	0.0132	0.1642	0.9362
X34	Leukopenia	−0.0575	0.1043	0.5823
X35	Band neutrophils	0.0035	0.0048	0.4746
X36	Neutrophils	−0.0172	0.0073	0.0195
X37	Neutrophilia	0.7804	0.1429	<0.0001
X38	Neutropenia	−0.1617	0.0890	0.0712
X39	Lymphocytes	0.0487	0.0260	0.0626
X40	Lymphopenia	0.0604	0.0931	0.5175
X41	C-reactive protein (CRP)	0.0007	0.0015	0.6506
X42	Elevated CRP	−0.1031	0.0764	0.1792
X43	Ferritin	−0.0004	0.0006	0.5288
X44	Prothrombin time	−0.0058	0.0373	0.8768
X45	Prothrombin time less than 12	−0.5818	0.3347	0.0840
X46	Prothrombin time more than 15	−0.4596	0.2747	0.0962
X47	Altered prothrombin time	0.7086	0.2572	0.0065
X48	Thrombin time	0.0021	0.0029	0.4837
X49	Activated partial thromboplastin time (aPTT)	0.0036	0.0033	0.2851
X50	aPTT more than 35 s	0.0532	0.0810	0.5121
X51	Fibrinogen	−0.0244	0.0770	0.7521
X52	Fibrinogen less than 2 g/L	0.0203	0.1270	0.8733
X53	Fibrinogen more than 4 g/L	−0.1141	0.1690	0.5007
X54	D-dimer, ng/mL (N < 250)	<0.0001	<0.0001	0.5400
X55	Vitamin D, ng/mL (N > 30)	0.0002	0.0022	0.9459
X56	Vitamin D deficiency	0.1064	0.1127	0.3464
X57	Zinc, μmol/L (Normal: 10.7–24.0)	0.0059	0.0056	0.2864
X58	Zinc deficiency	−0.0112	0.0889	0.8997

**Table 3 ijerph-22-01693-t003:** Coefficients of the first model taking into account non-hospitalized patients according to multivariate regression analysis for determining the risk factors for the DLCC, with the inclusion of 4 reliable predictors.

Conventional Designations of Factors in the Mathematical Forecasting Model	Name of Predictors	Regression Coefficient (B)	Standard Error, SE (B)	Significance Level *p*
X26	Obesity	0.4266	0.0949	<0.0001
X28	Allergic pathology	0.2855	0.0616	<0.0001
X33	Vitamin D level	0.0041	0.0018	0.0238
X34	Vitamin D deficiency	0.3972	0.0887	<0.0001

**Table 4 ijerph-22-01693-t004:** Coefficients of the second model taking into account non-hospitalized patients according to multivariate regression analysis for determining the risk factors for the DLCC, with the inclusion of 7 reliable predictors.

Conventional Designations of Factors in the Mathematical Forecasting Model	Name of Predictors	Regression Coefficient (B)	Standard Error, SE (B)	Significance Level *p*
X5	Fever	0.3018	0.1184	0.0117
X15	Number of symptoms ≥3	0.1579	0.0763	0.0400
X24	Allergic pathology	0.0600	0.0635	<0.0001
X30	Thrombocytosis	0.3383	0.0936	0.0004
X32	Leukocytes	−0.0307	0.0082	0.0002
X37	Neutrophilia	0.4646	0.1033	<0.0001
X47	Altered prothrombin time	0.1791	0.0582	0.0024

**Table 5 ijerph-22-01693-t005:** The result of the assessment of the acceptability of model 1, taking into account non-hospitalized patients, for predicting the risk of DLCC using ANOVA analysis.

Effect	Sums of Squares of Deviations	Degrees of Freedom	Mean Square Value	Fisher’s Exact Test	Significance Level *p*
Regression	13.25	4	3.31	16.29	<0.00001
Residual Deviations	52.89	260	0.2034		
General	66.14				

**Table 6 ijerph-22-01693-t006:** The result of the assessment of the acceptability of model 2, without taking into account non-hospitalized patients, for predicting the risk of DLCC using ANOVA analysis.

Effect	Sums of Squares of Deviations	Degrees of Freedom	Mean Square Value	Fisher’s Exact Test	Significance Level *p*
Regression	17.18	7	2.45	16.21	<0.00001
Residual Deviations	27.10	179	0.15		
General	44.29				

## Data Availability

All authors declare that they are willing to provide any documents supporting the reported results, including datasets analyzed or generated during the study, upon request.

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
