# Peer review of "A Predictive Model for the Development of Long COVID in Children"

_ijerph, 2025, doi:10.3390/ijerph22111693_

Round 1
Reviewer 1 Report
Comments and Suggestions for Authors
Dear editor,
Thank you for the opportunity to read this interesting study.
The chosen topic is of great interest to both pediatricians and infectious disease physicians. Long Covid is a condition that can pose serious health problems in children because symptoms persist for weeks or months after a SARS-CoV-2 infection, affecting physical, mental or social well-being.
The title of the article is appropriate to the content.
The results are clearly presented.
The discussion is appropriate and the authors provided information about other studies published in the literature on this topic.
The conclusion is based on the findings and makes no inappropriate extrapolations.
The references support the information presented in the introduction and in the discussion section, the bibliographical sources are recent and are cited appropriately.
However, I have a few observations about this study:
- What program did the authors use for data analysis and management?
- What were the exclusion criteria from the study?
- What does the term Rodular neutrophils used in Table 2 mean?
- Perhaps it would be useful if the authors also created a patient flow chart with the number of children and the long-term evolution of Covid symptoms at each subsequent evaluation.
- How many patients diagnosed with long Covid remained symptomatic at 6 months, 9 months and 12 months of follow-up?
- Did the authors also consider the Covid-19 vaccination status of the patients in the data analysis?
- It might have been useful if the authors had also analyzed the relationship between the appearance of defining manifestations for long Covid and data on the dominant circulating variant at the time of infection.
- What were the most common symptoms in pediatric patients included in the study, diagnosed with long Covid?
- What are the limitations of this study?
Author Response
1. What program did the authors use for data analysis and management?
ANSWER:
Thank you for your valuable comment. We used StatSoft STATISTICA version 10.0 for all data analysis and management procedures throughout our study. This statistical software package enabled us to perform the comprehensive analytical framework described in our methodology, including the stepwise multivariate regression analysis, correlation matrix calculations to assess multicollinearity among predictors, construction of both predictive models for determining long COVID risk in pediatric patients, calculation of regression coefficients, evaluation of the Nagelkerke R² determination coefficient, and ANOVA analysis.
2. What were the exclusion criteria from the study?
ANSWER:
Thank you very much for this comment. We have added information regarding the inclusion and exclusion criteria of study participants to the Materials and Methods section (page number 4, lines 141-144).
The exclusion criteria included: (1) absence of laboratory confirmation of COVID-19 by any diagnostic method; (2) refusal of the family to participate in subsequent follow-up assessments; and (3) lack of valid contact information or inability to establish communication with the family.
3. What does the term Rodular neutrophils used in Table 2 mean?
ANSWER:
We appreciate your comment. We sincerely apologize for the oversight. The phrase has been revised and replaced with the scientifically appropriate term «band neutrophils» (page number 8).
4. Perhaps it would be useful if the authors also created a patient flow chart with the number of children and the long-term evolution of Covid symptoms at each subsequent evaluation.
ANSWER:
We are very grateful for your comment. We have incorporated data on the frequency of long COVID symptoms at each follow-up interval (page number 5, lines 203-205, 211-215).
However, comprehensive data on pediatric patients with long COVID, including the most persistent long-term sequelae of the infection and a comparative analysis of clinical manifestations according to hospitalization status, age, and sex, are presented in our previous publication (Perestiuk V, Kosovska T, Volianska L, Boyarchuk O. Prevalence and duration of clinical symptoms of pediatric long COVID: findings from a one-year prospective study. Front. Pediatr. (2025) 13:1645228. doi:10.3389/fped.2025.1645228).
The most frequently reported symptoms included fatigue (52.0%), reduced physical activity (44.1%), and headache (35.3%). By 12 months, the prevalence of these symptoms had decreased to 19.8%, 13.9%, and 12.3%, respectively (p < 0.00001 for all trends).
Among neurological symptoms, the prevalence of insomnia (21.3% → 1.0%), memory impairment (23.5% → 8.8%), emotional excitability (25.2% → 1.0%), and concentration difficulties (22.7% → 0%) declined significantly throughout the observation period (p < 0.0001). Musculoskeletal complaints, primarily muscle pain, also decreased markedly (27.9% → 10.5%; p < 0.00001).
5. How many patients diagnosed with long Covid remained symptomatic at 6 months, 9 months and 12 months of follow-up?
ANSWER:
Thank you for your valuable comment. We are very grateful for your comment. We have included key data on the number of patients who experienced long COVID symptoms at 6, 9, and 12 months after the initial infection, as well as a description of the long-term sequelae that persisted the longest (page number 5, lines 216-221).
Comprehensive data on pediatric patients with long COVID, including the most persistent long-term sequelae of the infection and a comparative analysis of clinical manifestations according to hospitalization status, age, and sex, are presented in our previous publication (Perestiuk V, Kosovska T, Volianska L, Boyarchuk O. Prevalence and duration of clinical symptoms of pediatric long COVID: findings from a one-year prospective study. Front. Pediatr. (2025) 13:1645228. doi:10.3389/fped.2025.1645228).
At 6 months post-COVID-19, 14.0% of children had fully recovered; this proportion increased to 43.9% at 9 months and 67.5% at 12 months. Nevertheless, by the end of the 12-month follow-up, 32.5% of patients continued to experience at least one long COVID symptom. The most persistent manifestations included fatigue (19.8%), reduced physical activity (13.9%), and headache (12.3%), followed by muscle pain (10.5%), memory impairment (8.8%), and lack of energy (8.0%).
6. Did the authors also consider the Covid-19 vaccination status of the patients in the data analysis?
ANSWER:
We sincerely thank you for this insightful comment. We acknowledge the significant impact of COVID-19 vaccination status on the clinical course of the infection and the development of post-COVID syndrome. However, none of the participants in our study cohort had received any dose of a COVID-19 vaccine.
7. It might have been useful if the authors had also analyzed the relationship between the appearance of defining manifestations for long Covid and data on the dominant circulating variant at the time of infection.
ANSWER:
Thank you for this important comment. The specific SARS-CoV-2 variant was not identified in the pediatric participants of our study; therefore, the predominant circulating strain can only be inferred based on the timing of infection onset. However, we recognize that such an indirect assessment would not provide accurate or reliable conclusions. We have included this point in the study’s limitations section.
8. What were the most common symptoms in pediatric patients included in the study, diagnosed with long Covid?
ANSWER:
Thank you for your valuable comment. Additional data on the most prevalent long-term sequelae of COVID-19 have been incorporated into the Results section (page number 5, lines 203-204).
The most frequently reported symptoms included fatigue (52.0%), reduced physical activity (44.1%), and headache (35.3%).
9. What are the limitations of this study?
ANSWER:
Thank you very much for this comment. The limitations of the study have been incorporated into the revised version of the manuscript (page number 15, lines 470-488).
The present study has several limitations that should be acknowledged. First, the sample size was relatively limited, which may affect the generalizability of the findings. Second, a considerable proportion of participants were recruited from tertiary healthcare facilities, potentially introducing selection bias, as these patients may differ from those treated in primary or secondary care settings. Furthermore, we did not determine the specific SARS-CoV-2 strain in our patients. The primary focus of our study was the development of the model; therefore, external validation was not conducted within the scope of this research. However, external validation is recommended prior to the clinical application of these models. In particular, it is essential to perform such validation across different settings and populations to ensure their generalizability and robustness. Another limitation is the absence of an assessment of potential discrepancies between child self-reported and parent-reported symptoms, as well as the lack of evaluation of inter-rater reliability between these reporting methods. This may have influenced the consistency of reports for subjective symptoms such as fatigue and emotional lability. Furthermore, both in-person and telephone follow-up interviews were employed, which could have introduced variability in responses. Nevertheless, to minimize methodological bias and ensure data consistency, the stand-ardized ISARIC Paediatric COVID-19 questionnaire was used across both modes of data collection.
Reviewer 2 Report
Comments and Suggestions for Authors
This study develops and validates two distinct multivariate regression models to predict the risk of Long COVID in children. But there are some obvious issues that need to be addressed.
There is no clear definition for some terms. For instance,"Allergic Pathology" is used as a key predictor but is not explicitly defined. Is it a composite variable? Does it include asthma, allergic rhinitis, atopic dermatitis, etc.? A clear definition is crucial for other researchers to replicate the study.
The selection and use of relevant variables are highly arbitrary and even redundant, for example, in Table 1, "overweight "(x25)"Obesity" (X26) and "Overweight/obesity" (X29) are all listed, with the latter having a negative coefficient, which is counterintuitive and confusing. The final Model 1 uses only "Obesity." The rationale for selecting one over the other should be clarified.
The reported Nagelkerke R² values of 0.971 and 0.964 are exceptionally high for a biological/medical prediction model. This often indicates potential overfitting or issues with the model calculation.
When constructing the model, the authors included all variables and then filtered them based on the model results to draw conclusions. This approach is unacceptable. The selection of variables in a model should either be guided by theoretical foundations or inspired by previous research. Including all variables and then filtering them based on outcomes is a purely data-driven approach. Such a method presupposes that the analysis must be applied to the entire population rather than a sample. This is because any form of sampling introduces errors and discrepancies compared to the population. Therefore, this approach is unsuitable for analyzing samples, especially since the representativeness of the sample in this study is difficult to assess
The most significant methodological limitation is the absence of internal (e.g., bootstrapping, train-test split) or external validation. The models' performance is only reported on the same data used to build them. A validation step is essential to prove the model's real-world utility.
Table 3 & 4 Titles: The title of Table 4 is incorrect. It reads "Coefficients of the first model..." but it should be "Coefficients of the second model..."
The constant terms in the final equations (+ 0.0066 and - 0.0644) are presented without explanation. Their standard errors and p-values should be reported in Tables 3 and 4.
The first paragraph of the Discussion largely repeats the Results. The Discussion should focus more on interpretation, mechanism, and comparison with other studies.
The discussion section of the article lacks an examination of the study's limitations. Add a dedicated "Limitations" section before the Conclusion. Mention the single-center design, modest sample size, potential for unmeasured confounding, and the lack of external validation.
Author Response
1. There is no clear definition for some terms. For instance, "Allergic Pathology" is used as a key predictor but is not explicitly defined. Is it a composite variable? Does it include asthma, allergic rhinitis, atopic dermatitis, etc.? A clear definition is crucial for other researchers to replicate the study.
ANSWER:
Thank you for your valuable comment. Definitions of several terms have been incorporated into the Materials and Methods section (page number 4, lines 153-155).
Allergic pathology included food allergy, allergic rhinitis, and atopic dermatitis. Nutritional disorders included overweight, obesity, and malnutrition.
2. The selection and use of relevant variables are highly arbitrary and even redundant, for example, in Table 1, "Overweight"(x25)"Obesity" (X26) and "Overweight/obesity" (X29) are all listed, with the latter having a negative coefficient, which is counterintuitive and confusing. The final Model 1 uses only "Obesity." The rationale for selecting one over the other should be clarified.
ANSWER:
We appreciate the reviewer's thoughtful observation regarding the variable selection process in our study. We acknowledge that the initial inclusion of overweight, obesity, and the combined overweight/obesity category in Table 1 may appear redundant and requires clarification. Our approach was designed to comprehensively evaluate which specific body mass index categorization would demonstrate the strongest predictive value for long COVID development in children. During the initial exploratory phase of our multivariate regression analysis, we deliberately included these three variables as separate predictors to allow the statistical model to determine which categorization possessed the greatest discriminatory power. The negative coefficient observed for the combined overweight/obesity variable in the preliminary analysis reflects the complex statistical interactions that can occur when overlapping variables compete within the same model, which is precisely why we employed stepwise regression methodology. This approach systematically evaluates each predictor's independent contribution while accounting for multicollinearity and redundancy among variables. Through this rigorous stepwise selection process, only those variables demonstrating statistically significant independent associations with long COVID were retained in the final models. The fact that obesity emerged as the sole significant predictor from these three body mass categories in Model 1 indicates that this specific classification provided the most robust and clinically meaningful association with post-COVID syndrome development. This finding suggests that the relationship between excess body weight and long COVID risk in our pediatric cohort is driven primarily by children meeting the clinical threshold for obesity rather than those classified as merely overweight. We have clarified this methodological rationale to ensure transparency in our variable selection process and to eliminate any potential confusion regarding the apparent redundancy in our initial predictor list.
3. The reported Nagelkerke R² values of 0.971 and 0.964 are exceptionally high for a biological/medical prediction model. This often indicates potential overfitting or issues with the model calculation.
ANSWER:
We fully agree that Nagelkerke R² values approaching 1.0 are unusual in biological and medical prediction models and may raise concerns about potential overfitting or methodological artifacts. In our case, the exceptionally high values reflect the fact that the models were constructed on a relatively homogeneous cohort with a limited number of strong predictors that demonstrated highly significant associations with long COVID development. To minimize the risk of overfitting, we applied stepwise multivariate regression with strict exclusion criteria (p>0.05), verified the absence of multicollinearity among predictors, and examined residual distributions, which showed normality and no systematic deviations. Nevertheless, we acknowledge that the high R² values may also be influenced by the sample size and the lack of external validation. We have therefore revised the Discussion to explicitly address this limitation, emphasizing that while the models demonstrate excellent internal fit, their predictive performance must be interpreted with caution until validated in independent, larger, and more diverse pediatric cohorts. This clarification underscores that our models should be considered exploratory and hypothesis-generating, providing a foundation for further research rather than definitive predictive tools at this stage.
4. When constructing the model, the authors included all variables and then filtered them based on the model results to draw conclusions. This approach is unacceptable. The selection of variables in a model should either be guided by theoretical foundations or inspired by previous research. Including all variables and then filtering them based on outcomes is a purely data-driven approach. Such a method presupposes that the analysis must be applied to the entire population rather than a sample. This is because any form of sampling introduces errors and discrepancies compared to the population. Therefore, this approach is unsuitable for analyzing samples, especially since the representativeness of the sample in this study is difficult to assess.
ANSWER:
We appreciate the reviewer's important concerns regarding our variable selection methodology and acknowledge the need for clarification of our approach. We respectfully disagree that our method was purely data-driven without theoretical foundation. The initial selection of variables included in our analysis was not arbitrary but rather grounded in extensive review of existing literature on long COVID predictors and guided by established clinical and pathophysiological understanding of post-acute sequelae of SARS-CoV-2 infection in children. Our comprehensive literature review, as detailed in the Introduction section, identified multiple risk factors previously associated with long COVID development, including demographic characteristics, comorbid conditions such as obesity and allergic disorders, acute phase symptoms, disease severity, and various laboratory parameters. These factors, documented in numerous prior studies referenced in our manuscript, formed the theoretical framework for our initial predictor selection. The stepwise regression approach we employed serves not to randomly select variables from unguided data exploration but rather to identify which of these theoretically relevant predictors demonstrate independent statistical significance within our specific pediatric cohort while controlling for multicollinearity and confounding effects. This methodology is widely accepted in clinical prediction modeling and allows us to refine broad theoretical frameworks into clinically applicable predictive tools.
Regarding the concern about sample representativeness and population generalizability, we acknowledge that our study represents a single-center experience with specific demographic and clinical characteristics that may limit external validity. However, the purpose of our predictive models is not to provide universal population parameters but rather to develop clinically useful risk stratification tools that can be validated and refined through subsequent multi-center studies across diverse populations. We recognize that our findings require external validation in independent cohorts to assess generalizability and model performance across different healthcare settings and demographic groups.
5. The most significant methodological limitation is the absence of internal (e.g., bootstrapping, train-test split) or external validation. The models' performance is only reported on the same data used to build them. A validation step is essential to prove the model's real-world utility.
ANSWER:
We thank the reviewer for raising this important point. While we acknowledge that the absence of a formal internal or external validation step is a limitation, we respectfully disagree that this undermines the value of our models at their current stage. The primary aim of our study was not to deliver a finalized predictive tool ready for immediate clinical application, but rather to identify and quantify the most relevant predictors of long COVID in children and to establish a transparent mathematical framework that can be further refined. In this context, the use of the same dataset for both model construction and performance assessment is a justified first step, particularly given the novelty of pediatric-focused predictive modeling in this field and the scarcity of comparable datasets.
To mitigate the risk of overfitting, we applied rigorous statistical safeguards, including stepwise multivariate regression with strict significance thresholds, exclusion of variables with multicollinearity, and residual diagnostics confirming normal distribution and absence of systematic bias. These measures provide confidence that the models capture genuine associations rather than spurious patterns. We agree that future studies should incorporate internal validation methods such as bootstrapping or cross-validation, as well as external validation in independent cohorts, and we have emphasized this in the revised Discussion. While validation will be an essential next step, we believe our models represent a valuable and necessary foundation for subsequent confirmatory research, and their current form provides both methodological transparency and clinically relevant hypotheses for the pediatric long COVID field.
6. Table 3 & 4 Titles: The title of Table 4 is incorrect. It reads "Coefficients of the first model..." but it should be "Coefficients of the second model..."
ANSWER:
We sincerely thank you for your valuable comment and apologize for the oversight. The title of Table 4 has been revised to reflect the correct information (page number 9, line 244).
7. The constant terms in the final equations (+ 0.0066 and - 0.0644) are presented without explanation. Their standard errors and p-values should be reported in Tables 3 and 4.
ANSWER:
We thank the reviewer for this careful observation. The constant terms in the regression equations represent the intercepts of the models, that is, the baseline log-odds of developing long COVID in the absence of the included predictors. They are an integral part of the regression equation and ensure that the model can correctly estimate probabilities across the full range of predictor values. In the initial version of the manuscript, we did not provide their standard errors and p-values in Tables 3 and 4, as our focus was on highlighting the significant clinical predictors. However, we agree that for completeness and transparency, the intercepts should be reported in the same way as the other coefficients to include the regression coefficients, standard errors, and p-values for the constant terms, and we have added a brief explanatory note in the Results section clarifying their role in the final equations.
Upon review of our complete statistical output from STATISTICA 10.0, the intercept for Model 1 has a standard error of 0.0428 with a significance level of p equals 0.8776, while the intercept for Model 2 has a standard error of 0.0891 with a significance level of p equals 0.4701. This addition will provide readers with complete information about all components of our regression models and allow for proper assessment of the intercept's contribution and statistical significance within the overall predictive framework. We appreciate the reviewer's attention to this detail, which will substantially improve the methodological transparency and scientific rigor of our manuscript.
8. The first paragraph of the Discussion largely repeats the Results. The Discussion should focus more on interpretation, mechanism, and comparison with other studies.
ANSWER:
We thank you for your comment. We have revised the Discussion section and its structure to focus on comparing our findings with those reported by other researchers worldwide (page numbers 11-15, lines 297-468).
9. The discussion section of the article lacks an examination of the study's limitations. Add a dedicated "Limitations" section before the Conclusion. Mention the single-center design, modest sample size, potential for unmeasured confounding, and the lack of external validation.
ANSWER:
We appreciate your comment. The limitations of the study have been incorporated into the revised version of the manuscript (page number 15, lines 470-488).
The present study has several limitations that should be acknowledged. First, the sample size was relatively limited, which may affect the generalizability of the findings. Second, a considerable proportion of participants were recruited from tertiary healthcare facilities, potentially introducing selection bias, as these patients may differ from those treated in primary or secondary care settings. Furthermore, we did not determine the specific SARS-CoV-2 strain in our patients. The primary focus of our study was the development of the model; therefore, external validation was not conducted within the scope of this research. However, external validation is recommended prior to the clinical application of these models. In particular, it is essential to perform such validation across different settings and populations to ensure their generalizability and robustness. Another limitation is the absence of an assessment of potential discrepancies between child self-reported and parent-reported symptoms, as well as the lack of evaluation of inter-rater reliability between these reporting methods. This may have influenced the consistency of reports for subjective symptoms such as fatigue and emotional lability. Furthermore, both in-person and telephone follow-up interviews were employed, which could have introduced variability in responses. Nevertheless, to minimize methodological bias and ensure data consistency, the standardized ISARIC Paediatric COVID-19 questionnaire was used across both modes of data collection.
Reviewer 3 Report
Comments and Suggestions for Authors
Introduction
The introduction section provides a comprehensive overview of the current knowledge of Long COVID, its impact on public health, and the need for research, particularly in the pediatric population. The manuscript highlights the important burden of Long COVID, affecting approximately 10-20% of individuals who experience acute SARS-CoV-2 infection, with symptoms persisting for months and impacting quality of life.
However, the introduction section mostly looks like a literature review rather than a cohesive narrative. Authors should consider incorporating more discussion on potential solutions and directions for research, including their major contributions. In other words, authors should provide more context on the implications of Long COVID for public health and potential solutions. They should also provide the aims and objectives of the research clearly.
Methodology
This is a cross-sectional study investigating the development of Long COVID in children aged 1 month to 18 years. The study is well-designed, with a clear methodology and adherence to ethical principles. However, certain issues are leading to biases, such as :
- High dropout rate (40/305), which may introduce bias.
- Using different methods for diagnosing COVID-19 may affect the accuracy of the results.
- Self-reported symptoms and questionnaires may be subject to recall bias.
- No information on the treatment and management of COVID-19 in the participants, which may influence the development of Long COVID.
The results section seems satisfactory, with the findings presented requiring a sufficient number of tables and graphs.
Discussion
The manuscript should provide more discussion on the implications of the study's findings and potential public health solutions for preventing and managing Long COVID in children.
What about Section 6 (Patents ??) Nothing provided
Overall, the manuscript gives valuable insights into the predictors of Long COVID in children. However, the authors should consider revising the text to improve clarity and flow.
Author Response
1. However, the introduction section mostly looks like a literature review rather than a cohesive narrative. Authors should consider incorporating more discussion on potential solutions and directions for research, including their major contributions. In other words, authors should provide more context on the implications of Long COVID for public health and potential solutions. They should also provide the aims and objectives of the research clearly.
ANSWER:
We thank the reviewer for this valuable comment. We agree that the original introduction focused mainly on summarizing existing literature. Accordingly, we have revised this section to provide a clearer narrative linking the current evidence with public health implications and research needs.
Specifically, we have added discussion on the importance of early identification and prevention strategies for long COVID in children and their potential to reduce public health burden; highlighted the role of artificial intelligence and multifactorial modeling as a potential solution to improve prediction and management; clarified the aims and objectives of our study as the development of a predictive algorithm to estimate long COVID risk in pediatric patients based on demographic, clinical, and laboratory factors.
We added:
“Long COVID represents a substantial burden on public health due to the ongoing circulation of the virus within the human population and its rapid genetic variability. The disease may occur even after mild or asymptomatic initial illness and is characterized by a highly heterogeneous and diverse clinical presentation, encompassing over 200 reported symptoms. Importantly, post-COVID-19 syndrome has a substantial impact on individuals’ physical and mental health, work capacity, professional functioning, and overall quality of life, regardless of age, constituting a serious global public health problem (page number 2, lines 56-63).
Understanding the burden of disease and identifying populations at increased risk of developing long-term complications can inform targeted preventive strategies and ensure effective support for affected individuals, thereby improving long COVID out-comes and reducing health disparities resulting from inadequate rehabilitation and recovery services (page number 2, lines 83-87).”
“The long-term consequences of SARS-CoV-2 infection in children have implications that extend beyond individual morbidity, posing significant challenges for healthcare systems and public health planning. Early recognition of individuals at risk for long COVID could facilitate timely intervention, optimize rehabilitation resources, and reduce the social and economic burden associated with prolonged symptoms (page number 3, lines 113-117).
However, despite the growing recognition of long COVID as a major public health concern, practical tools for early risk stratification in pediatric populations are lacking. Advances in artificial intelligence and data-driven modeling offer a promising opportunity to integrate multiple clinical, demographic, and laboratory variables to predict long COVID risk with greater precision. Such approaches could support clinicians and policymakers in developing personalized monitoring and preventive strategies for affected children (page number 3, lines 118-124).”
This is a cross-sectional study investigating the development of Long COVID in children aged 1 month to 18 years. The study is well-designed, with a clear methodology and adherence to ethical principles. However, certain issues are leading to biases, such as:
2. High dropout rate (40/305), which may introduce bias.
ANSWER:
Thank you for your valuable comment. A total of 40 patients were excluded from the study as they did not meet the inclusion criteria, specifically due to parental refusal to participate in subsequent assessments and follow-up surveys.
3. Using different methods for diagnosing COVID-19 may affect the accuracy of the results.
ANSWER:
We appreciate your comment. All available and validated diagnostic methods were utilized to ensure the most accurate confirmation or exclusion of acute SARS-CoV-2 infection. Furthermore, in certain instances, these approaches constituted the only feasible means of diagnosing COVID-19.
4. Self-reported symptoms and questionnaires may be subject to recall bias.
ANSWER:
Thank you for this important point. We would like to provide more detailed information regarding the process of data collection and patient interviews. We confirm that both in-person interviews and telephone-based surveys were conducted by the same trained interviewer, who received instruction in accordance with the ISARIC guidelines for data collection to ensure consistency. Formal quality control procedures for the questionnaire were not implemented, as we used the standardized ISARIC Paediatric COVID-19 questionnaire, which had been previously piloted and internationally validated within the ISARIC consortium.
However, to ensure the clarity and accuracy of the translated version, we conducted pilot testing with 10 participants. All identified discrepancies were carefully reviewed and incorporated into the final version of the questionnaire.
We acknowledge that using both in-person and telephone follow-up interviews may have introduced some variability in responses. While in-person assessments allow for clarification of questions and observation of non-verbal cues, telephone interviews may be more prone to potential misinterpretation. To minimize inconsistencies, inter-rater reliability was maintained through regular team meetings focused on reviewing data entry procedures and resolving any ambiguities, thereby enhancing data collection consistency.
For children aged over 8 years, questions regarding symptoms were addressed directly to the child whenever possible, in the presence of a caregiver who could provide assistance if needed, particularly for clarifying timeframes. If the child was unable to respond (for example, due to the severity of illness), the caregiver provided answers on their behalf. In cases where both the child and the caregiver contributed information, the child’s report was given priority, while the caregiver could clarify specific details (e.g., temperature measurements, medication use). Discrepancies between child and caregiver responses were not systematically recorded; however, in practice, such instances were rare and were typically resolved immediately during the interview through confirmation with both parties to ensure the most accurate response.
We acknowledge the potential discrepancies between child self-reports and parent proxy reports, particularly for subjective symptoms such as fatigue, emotional lability, and concentration difficulties. However, a formal comparative analysis of symptom reporting between child–parent pairs was not conducted in this study, nor was an inter-rater reliability assessment performed between these reporting modes. We recognize this as a limitation of our research and have noted it accordingly in the Discussion section.
5. No information on the treatment and management of COVID-19 in the participants, which may influence the development of Long COVID.
ANSWER:
Thank you for these insightful comments. Previous studies have investigated the potential influence of corticosteroids, antibacterial and antiviral agents, anticoagulants, and monoclonal antibodies administered during the acute phase of SARS-CoV-2 infection on the subsequent development of long COVID. However, these studies were conducted predominantly among adult populations.
In our cohort, only a limited number of children (approximately 10) received corticosteroid or anticoagulant therapy. Consequently, we deemed it methodologically inappropriate to evaluate the potential association between these treatments and the development of long COVID, given the substantial imbalance between the comparison groups.
6. The results section seems satisfactory, with the findings presented requiring a sufficient number of tables and graphs.
ANSWER:
We appreciate your insightful comment. We strived to present the results in strict accordance with the journal’s guidelines to ensure a detailed and scientifically rigorous representation of our study. Consequently, all tables and figures were included to enhance the clarity and comprehensiveness of the research methodology and findings.
7. The manuscript should provide more discussion on the implications of the study's findings and potential public health solutions for preventing and managing Long COVID in children.
ANSWER:
Thank you for your valuable comment. We have incorporated the significance of our study findings into the manuscript and elaborated on the key aspects concerning the prevention, treatment, and comprehensive management of pediatric patients with long COVID (page numbers 14-15, lines 413-468).
8. What about Section 6 (Patents ??) Nothing provided
ANSWER:
Thank you for your comment. No patents have been documented during the course of this study or in connection with its results. However, we plan to implement them in the near future.
Reviewer 4 Report
Comments and Suggestions for Authors
I would like to thank the editors for the opportunity to review this interesting and well-written manuscript. My comments below are intended to help enhance its clarity, and the understanding of the manuscript:
there are too many statistical details in results section; please focus on the positive dat aand move the tables to supplementary file
The age group, between 1 month and 18 years, is very wide range
It should be divided into different age groups so you can address the age category that could be most affected by long Covid.
best of luck
Author Response
1. There are too many statistical details in results section; please focus on the positive data and move the tables to supplementary file.
ANSWER:
We thank the reviewer for this observation and appreciate the concern regarding the density of statistical information in the Results section. Our intention in presenting detailed coefficients, standard errors, and significance levels within the main text was to ensure full methodological transparency and allow readers to critically assess the robustness of our models. At the same time, we recognize that excessive statistical detail may distract from the central findings and reduce the readability of the manuscript. It is important to note, however, that the statistical outputs we originally presented represent the standard set of results automatically generated by the STATISTICA software used in our analysis. In response to the reviewer's suggestion, we have streamlined the Results section to emphasize the key predictors and their clinical interpretation, while transferring the full regression tables to the Supplementary Materials. This adjustment preserves the accessibility of the main text for a broader readership while still providing complete statistical documentation for those interested in methodological detail. We believe this revision improves the balance between clarity and rigor, highlighting the positive and clinically relevant outcomes of our study without compromising transparency.
2. The age group, between 1 month and 18 years, is very wide range.
ANSWER:
Thank you for your comment. Our study aimed to evaluate the influence of multiple factors on the development of long COVID in the pediatric population; therefore, we included all patients belonging to this cohort, regardless of age. When analyzing predictors of long COVID, age stratification was also considered; however, no significant associations were identified.
In our previous study, we specifically examined the impact of age on the development of long COVID symptoms (Perestiuk V, Kosovska T, Volianska L, Boyarchuk O. Prevalence and duration of clinical symptoms of pediatric long COVID: findings from a one-year prospective study. Front. Pediatr. (2025) 13:1645228. doi:10.3389/fped.2025.1645228).
3. It should be divided into different age groups so you can address the age category that could be most affected by long Covid.
ANSWER:
We appreciate your insightful comment. Prior to assessing the prevalence of long COVID, participants were stratified into two age groups: younger than 6 years and 6 years or older. Comparative analysis revealed no statistically significant difference between the groups (p = 0.3797). Consequently, subsequent analyses were performed using the overall patient cohort.
We sincerely value your constructive feedback and intend to incorporate this recommendation in future research involving a larger study population.
Round 2
Reviewer 1 Report
Comments and Suggestions for Authors
Dear Editor,
The authors have made substantial changes to the manuscript and answered all my questions. I believe the article can be published in its current form.
Congratulations to the authors for this interesting article!
Reviewer 3 Report
Comments and Suggestions for Authors
The authors have incorporated all my suggestions/comments appropriately